

# Design of innovation ability evaluation model based on IPSO-LSTM in intelligent teaching

Fei Wan

School of Education, Guangzhou University, Guangzhou, Guangdong, China

## ABSTRACT

Guided by the development of an innovative economy, students' innovative education has also become the focus of talent training. This research aims to realize the intelligent evaluation of students' innovation ability. In this article, we proposed an innovation ability framework that integrates students' psychological state and innovation evaluation indicators. Firstly, the qualitative description of psychological data is quantified using the Delphi method. Secondly, this article proposes an improved particle swarm optimization-long short-term memory (IPSO-LSTM) model to achieve high-precision evaluation and classification of innovation capabilities. The classification accuracy of this model for excellent, general and failed innovation capabilities is up to 95.3%. Finally, the characteristic contribution analysis of psychological and innovative ability characteristics is carried out. The results show that the evaluation of creative ability contributes more than 50% to the psychological aspects of excellent students. This shows the importance of psychological status on creative ability and provides a theoretical basis for integrating innovative education and psychological education in the future.

# INTRODUCTION

As the 21st century ushered in the era of the knowledge economy, societal progress shifted from industrialization and information development to a focus on education and scientific research. Amidst global economic integration, the development of high-tech industries has surged with unprecedented speed. These industries have emerged as crucial drivers in the global economy, owing to their inherent characteristics such as high knowledge and technology content, low resource consumption, and substantial value addition. They have become the mainstay of economic advancement for all nations and a pivotal factor in the competition for economic advantages (*Schmidt, Schitter & Rankers, 2020*).

A country's core competitiveness will increasingly hinge on its capacity for independent innovation in this world, marked by intensifying scientific and technological competition. Consequently, expedited technological progress and industrial restructuring are essential to secure a competitive edge in the global arena. The vigorous development of high-tech industries is paramount to bolstering national competitiveness (*Davari et al., 2022*). The pivotal source and driving force behind industrial development are rooted in technological innovation ability and activities. Therefore, higher education and basic education must

Corresponding author
Fei Wan, 13802820756@163.com

prioritize cultivating students' innovation prowess to forge ahead in this dynamic landscape. Innovative education plays an extremely important role in today's fiercely competitive society. Firstly, from an economic perspective, innovative education is considered a key driving force for economic growth and social progress. Students who cultivate innovative abilities are more likely to become future entrepreneurs, innovation leaders, and problem solvers, promoting the development of emerging industries, creating job opportunities, and improving national competitiveness. Secondly, innovative education plays an important role in the field of knowledge. In the information age, knowledge acquisition and dissemination speed is unprecedented. Innovation education not only focuses on imparting knowledge but also cultivates students' ability to integrate, analyze, and create information, enabling them to adapt to constantly changing knowledge needs. In addition, innovative education also has social impacts. It emphasizes students' creative thinking, teamwork, communication skills, and problem-solving abilities, which are crucial in solving complex real-world challenges. Innovative education helps cultivate citizens' sense of social responsibility and can positively contribute to social and environmental sustainability.

The wellspring of innovation lies in cultivating talented individuals with innovative capabilities, a task best accomplished within a nurturing and innovative educational environment. Teachers who have received comprehensive innovation education are more likely to exhibit innovative behaviours during their teaching practices and skillfully employ various approaches to stimulate students' creative thinking, thus accomplishing the goal of fostering innovative minds (*Osmolovskaya, Ivanova & Klarin, 2019*). Therefore, to effectively nurture innovative talents, the journey must commence with the daily education of these individuals. This includes establishing a comprehensive evaluation system for innovation ability and directing particular attention to the development of students exhibiting strong innovation potential. In recent years, researchers have discovered a significant correlation between innovation psychological quality and innovation ability. The study of innovation psychological quality has evolved from focusing on specific aspects, such as thinking patterns, to embracing a holistic approach, viewing it as a comprehensive system (*Rahmat, Leng & Mashudi, 2020*). Initially, psychologists concentrated on the creative subject's innovation process or certain facets. However, through thorough and meticulous research, they have recognized that numerous factors impact the subject's capacity for innovation. Elements such as innovation awareness, innovation ability, competitiveness, and other aspects are equally pivotal factors (*Wang, Sun & Wu, 2021*).

Consequently, students' innovative abilities from their psychological and cognitive perspectives. However, due to the vast diversity in their performance, it becomes challenging to express their psychological and and cognitive state through simplistic tests. A method is urgently needed to quantitatively assess students' psychological and cognitive state and intelligently evaluate their innovative capacity. To address this, this article proposes an innovation ability evaluation model that combines students' psychological cognition level with the characteristics of both psychological and innovation evaluations.

This model is a valuable reference for future assessments of students' innovation capabilities. The contributions of this article can be summarized as follows:

1) Employing the Delphi method, the article quantifies the qualitative description of psychological data, ensuring a robust and rigorous approach to psychological evaluation.

2) The article introduces an innovative IPSO-LSTM method that effectively integrates the quantitative data of psychological attributes with innovation survey data. This fusion allows for the intelligent classification of students into categories of excellent innovation ability, general ability, and failure ability. The model exhibits an impressive precision rate of 95.3% on the test set.

3) After completing students' innovation ability evaluation, the article analyses the contribution rates of psychological and innovation characteristics. Remarkably, it is revealed that the contribution rate of psychological characteristics among students with excellent innovation ability exceeds 50%, underscoring the significant impact of psychological status on innovation prowess.

The rest of this article is organized as follows: "Related Works" introduces related works for innovation model establishment; in "Innovation Evaluation Model Establishment Using IPSO-LSTM", Delphi, LSTM and IPSO are introduced to construct the IPSO-LSTM model; "Experiments and Analysis" describes the experiment and result analysis of the innovation ability classification; In "Discussion", the author discusses the result and what should be paid for future innovation education; the conclusion is presented in "Conclusion".

# RELATED WORKS

## Evaluation model of innovation capability

Innovation capability is a multifaceted construct that necessitates consideration from various perspectives. In domestic and international research, the academic community has enhanced the evaluation model of innovation capability by constructing pertinent indicators and assessment methods. For instance, *Romijn & Albaladejo (2002)* studied British software companies, investigating the determinants of their innovation capability. The study combined experimental innovation indices with traditional innovation performance indicators, uncovering the critical role of research and development and regional scientific bases in fostering high-tech derivatives. *Türker (2012)* devised a model to measure technological innovation capability in the context of automobile enterprises. The model encompassed "input factors," "process factors," and "output factors," considering the quantity and quality of supporting factors required for implementing technological innovation in enterprises. Building upon process and system theories, researchers regarded enterprise technological innovation as a systematic and continuous improvement process. They established an evaluation model of enterprise technological innovation capability from a process perspective. The model, tailored to the cement industry, encompassed innovation management, investment, research and development,

manufacturing, marketing promotion, and innovation output capacity (*Xu & Hua, 2012*). Researchers employed qualitative and quantitative methods to recognize the phased characteristics of the enterprise's technological innovation process and the allocation and utilization of innovation resources at each stage. They formulated 18 secondary indicators under six primary indicators, including collective learning, resources, marketing, innovation organization, strategic planning, and performance, to develop an evaluation framework for the technological innovation capability of research and technological organizations (*Ravari, 2016*). In sum, these studies exemplify the ongoing efforts within the academic community to enrich the evaluation models of innovation capability, encompassing diverse industries and focusing on distinct dimensions of innovation-related factors.

In evaluating innovation capability, researchers have adopted various methods to assess and analyze the innovation performance of different entities. *Wang & Huang (2007)* applied Data Envelopment Analysis (DEA) to evaluate R&D activities in other countries. The inputs considered were R&D capital stock and human resources, while the outputs were patents and academic publications. *Sumrit & Anuntavoranich (2012)* constructed a comprehensive evaluation index system for enterprises' technological innovation capability, encompassing six levels: management, collective learning, innovation procurement, technology development, process design, and technology commercialization. They analyzed this system using the Decision Test and Evaluation Experiment (DEMATEL) method, and the findings indicated that innovation management capability significantly influenced enterprises' technological innovation capability. Researchers employed the Analytic Hierarchy Process (AHP) to assign weights to each main index in their evaluation model. They further used fuzzy set theory to express fuzzy indices with numerical values, allowing for a more nuanced and flexible evaluation process (*Cho et al., 2015*). *Zhen & Yao*'s *(2021)* research used deep learning and neural network technology to evaluate enterprises' technological innovation capability comprehensively. This approach leveraged advanced computational methods to derive insightful assessments.

These diverse evaluation methods demonstrate the ongoing efforts to adopt sophisticated analytical techniques and leverage cutting-edge technologies to comprehensively assess innovation capability across various entities and industries. *Gruber (1988)* believes that forming innovation ability includes individual innovation motivation, mastery of conceptual principles, practical application skills, and psychological drive. *Sternberg & Lubart (1993)* proposed the famous three-dimensional model theory of innovation ability through the implicit theory analysis method of innovation ability, which mainly refers to the three dimensions of personality, intelligence, and cognitive style and is divided into six factors: intellectual process, knowledge, cognitive style, personality characteristics, motivation, and environment. Through the evaluation of innovation ability, psychologists' definition of innovation, and current innovation education, it is not difficult to see that the evaluation of innovation ability cannot focus on psychological factors. More psychological data must be introduced in innovation education for more detailed and accurate assessment.

## Model evaluation based on the artificial intelligence method

Artificial intelligence-based evaluation methods have widespread applications in current research across various fields. These methods have become the focal point of investigation, aiming to achieve quantitative evaluation of indicators and enhance predictive and evaluative accuracy by integrating machine learning and deep learning techniques. In the context of the rising popularity of online courses due to the COVID-19 pandemic, researchers have made notable strides in improving the reliability of models for predicting student behaviour and dropout rates in online learning environments. *Jin (2020)* proposed a support vector regression (SVR) model, utilizing an improved quantum particle swarm optimization (PSO) algorithm to extract weekly behaviour characteristics from students' online behaviour data, subsequently predicting abnormal learning statuses. To enhance the reliability of artificial neural network models, *Wang, Yu & Miao (2017)* selected 186 characteristic fields from the original data and integrated convolutional neural networks and cyclic neural networks to predict dropout. Similarly, *Feng, Tang & Liu (2019)* combined learner and course information with four learning behaviour record data types, achieving dropout probability prediction using deep neural networks (DNN). *Zhou & Xu (2020)* proposed a dropout prediction method that employed multi-model stack integrated learning.

Researchers have also explored novel methods to improve the accuracy of neural network models. *Jin (2021)* studied the calculation and implementation algorithm of the initial weights of each student, resulting in a significant enhancement in the prediction performance of the weighted training samples, surpassing the conventional random selection of initial values. *Fu et al. (2021)* utilized static attention to obtain attention weights on each dimension, improving model performance. In the context of online platforms like Coursera, *Edalati, Imran & Kastrati (2021)* employed machine learning algorithms to identify teaching-related content and predict students' emotional attitudes based on manually marked student comments. Remarkably, they achieved high performance, with the random forest model's F1 score for emotional classification reaching as high as 99.43%. Additionally, researchers have developed personalized linear multiple regression models to predict students' grades in real-time using data obtained from various MOOC platforms (*Ren, Rangwala & Johri, 2016*). *Yu, Wu & Liu (2019)* established a series of learning behaviour data based on students' video clickstreams and built a prediction model for learning outcomes, with the artificial neural network algorithm yielding the highest prediction accuracy among KNN, SVM, and NN approaches.

These studies illustrate the growing potential and utility of artificial intelligence techniques in various aspects of educational research and predictive modelling in the context of online learning platforms. In examining evaluation metrics and methodologies concerning innovation prowess, numerous scholars aspire to operationalize the intangible aspects of innovation appraisal and assess its capability through a multidimensional approach. Nevertheless, these evaluative endeavours necessitate substantial investment in model construction and quantification. In light of this, the methodology founded upon

machine learning and deep learning emerges as the foremost recourse to address such challenges. Consequently, the crux of evaluating innovation capability lies in the adept selection of pertinent features and establishing high-precision models.

# INNOVATION EVALUATION MODEL ESTABLISHMENT USING IPSO-LSTM

## Processing of evaluation indicators

Owing to the elusive nature of diverse psychological indicators, quantitative analysis through mathematical models becomes imperative for the subsequent evaluation of psychological cognition data. In this regard, the Delphi method is instrumental in surmounting the challenges associated with conducting such quantitative analyses, particularly in the evaluation domain, where indicator importance is pivotal in fostering information exchange and achieving consensus.

Delphi is a consultative approach within modern education evaluation, leveraging expert opinions to formulate an education evaluation plan and garner unanimous recognition for the significance of specific indicators. This method adopts a distribution question table to collect and collate input from experienced professionals, thereby statistically assessing the importance of indicators and diverse judgmental perspectives, culminating in a consensus-driven analysis of the pertinent issue. The process of this approach is visually depicted in the accompanying Fig. 1.

Therefore, based on the Delphi method, the quantitative analysis of students' psychological cognition level is completed, and relevant data indicators are formed and sent to the subsequent evaluation model. The specific steps are given as follows:

Although Delphi method has certain advantages in prediction and decision-making, it also has some drawbacks. Firstly, it relies on the subjective judgment of experts, making it susceptible to individual biases and errors, especially when there are differences of opinion among experts. Secondly, the Delphi method requires a significant amount of time and resources, as it typically involves multiple rounds of repeated expert investigations and feedback, which may not be suitable for emergency decision-making or situations with limited resources. In addition, the results of the Delphi method may not always be accurate, as it cannot eliminate uncertainty, and its predicted results may be influenced by the way the problem is stated and the survey design. Therefore, we use multiple iterations and external experts to make the data as accurate as possible and eliminate bias.

## The structure of the LSTM model

The innovation capability assessment transforms into a classic multi-source regression problem upon data quantification. Traditional linear models and machine learning techniques, like linear and logistic regression, exhibit limited efficacy in handling heterogeneous data. While classical methods such as random forest and SVM may enhance accuracy to some extent through feature transformations like Gaussian functions, their performance remains suboptimal (*Nasa, Jain & Juneja, 2021*).

The burgeoning computational power of computers has led to increased attention toward neural network-based models. This article adopts the long short-term memory

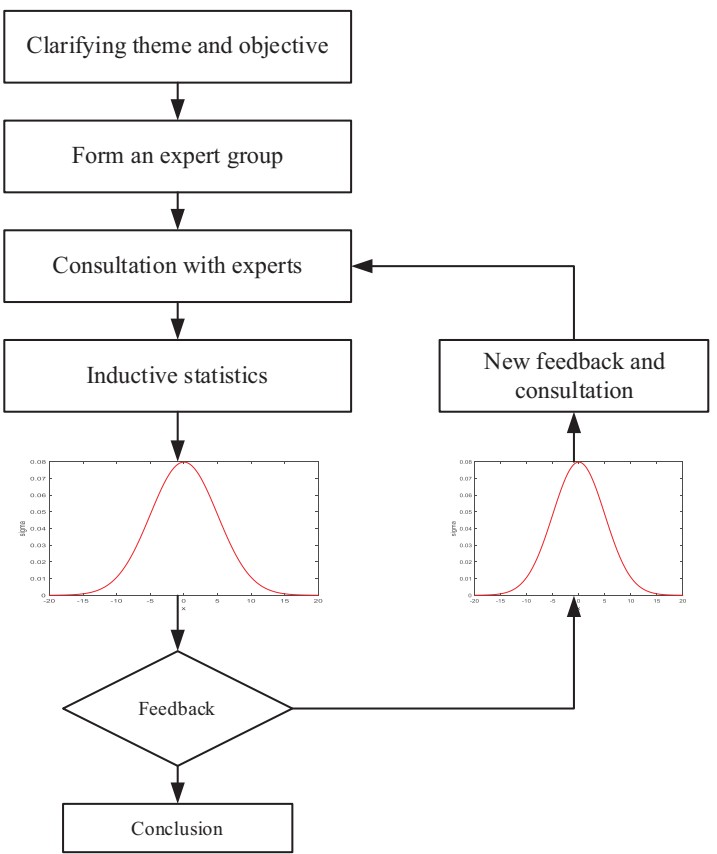

**Figure 1** **The framework for the Delphi.**

(LSTM) method as a foundational framework for the innovation capability evaluation model. By incorporating a "memory unit," LSTM surpasses recurrent neural network (RNN) in delivering superior results. As a sequence classification technique, LSTM is paramount in fully considering the interrelationships among various inputs, which is critical for the ultimate evaluation (*Shi et al., 2022*). The computation process of the LSTM method is expounded below:

$$f_t = \sigma(W_f \cdot [h_{t-1}, x_t] + b_f) \tag{1}$$

$$i_t = \sigma(W_i \cdot [h_{t-1}, x_t] + b_i) \tag{2}$$

$$o_t = \sigma(W_o[h_{t-1}, x_t] + b_o) \tag{3}$$

$$h_t = o_t \times \tanh(C_t) \tag{4}$$

Among them, $f_t$, $i_t$, and $o_t$ in Formulas (1)–(3) are forgetting gates, memory gates and output gates, respectively. These three special devices enable LSTM to achieve high-performance data classification and prediction tasks while completing the memory of each state in the sequence.

**Steps for the data quantification using the Delphi method.**

1. Problem determination: Select common psychological scales for psychological testing, and provide quantitative indicators based on students' innovation ability exams and practical scores.

2. Expert screening of effective scales and comprehensive evaluation of innovation ability interview scores.

3. First round of investigation and data organization: Re-collect data among classmates based on confirmed indicators;

4. Data feedback: Invite external experts to conduct preliminary evaluations of the model;

5. Data organization for the second round of survey: collect data from classmates again based on the confirmed indicators;

6. Repeat step 4 until 80% of experts are satisfied.

7. Summarize and analyze the truth setting of the input and output values of the model.

For the LSTM model, selecting the activation and loss functions will also greatly impact its final results. Commonly used activation functions contain sigmoid, tanh and softmax functions. Because the innovation evaluation process is not a simple two-classification process but is evaluated according to the grade, the Softmax function, as the extension of the logic function, has been widely used in different probability classification methods. Softmax function can also improve classification accuracy and support multi-classification. Therefore, this article selects the Softmax function as the activation function, and its specific calculation method is shown in Formula (5):

$$f(x) = \frac{e^{x_i}}{\sum_{j=1}^{n} e^{x_j}} \tag{5}$$

In addition to carefully selecting the activation function, selecting the loss function is also important. The loss function can highly affect the recognition ability of machine learning features. Cross entropy loss commonly plays a key role in training neural networks, which can determine the model performance. The calculation methods of the logarithmic loss function and cross-entropy loss function are shown in Eqs. (6) and (7), where y is the true value, $\hat{y}$ is the predicted value.

$$J(Q) = \frac{1}{m} \left[ \sum_{i=1}^{m} \log \hat{y} + (1 - y)\log(1 - \hat{y}) \right] \tag{6}$$

$$J(Q) = -\frac{1}{m} \sum_{i} [\hat{y} \ln a + (1 - \hat{y})\ln(1 - y)] \tag{7}$$

The proposed evaluation model of innovation capability, rooted in students' psychological cognition, fundamentally constitutes a multi-classification problem. The logarithmic loss function is typically employed for two-class classification tasks in such scenarios. However, to effectively handle the multi-classification nature of this study, the cross-entropy loss function is the preferred choice. Consequently, in the model construction using LSTM, this article adopts the cross-entropy function depicted in Formula (7).

## IPSO optimization for the LSTM model

Neural network methods often encounter the issue of converging to local optimal solutions due to the initial value setting, thereby impacting the model's accuracy. Hence, optimizing model parameters through intelligent algorithms has become a crucial approach to enhancing accuracy. This article employs the improved particle swarm optimization (IPSO) method to ameliorate algorithmic precision. IPSO represents an enhancement over the traditional PSO (*Chen, Xiong & Li, 2022*).

PSO draws inspiration from the behaviour of a group of birds searching for food in space. Particles within the algorithm adjust their positions based on individual and collective experiences to find the optimal solution. The specific realization process of PSO is depicted through Eqs. (8) and (9):

$$v_{i,t+1} = w^* v_{i,t} + c_1 * \text{rand} \,^* \left( \text{pbest}_1 - x_{i,t} \right) + c_2 * \text{rand} \,^* \left( \text{gbest}_1 - x_{i,t} \right) \tag{8}$$

$$x_{i,t+1} = x_{i,t} + \lambda * v_{i,t+1} \tag{9}$$

Despite the advantages of the PSO algorithm, including its relatively simple structure, limited parameter changes, and robustness, it suffers from reduced convergence efficiency in the later iterations. It is susceptible to getting trapped in local optima. Referring to Eqs. (8) and (9), improving convergence speed can be achieved by adjusting the learning factors c1 and c2. A larger value for these factors would increase the algorithm's step size but at the cost of decreased overall optimization efficiency. Similarly, to avoid local optima, one can increase the particle number update rate through the inertia weight w, but this would also lead to a decrease in the overall convergence speed of the algorithm.

This article optimizes the PSO method to address these challenges by introducing weight optimization and adding adaptive mutation particles. The IPSO method primarily focuses on enhancing the following two aspects:

(1) Change inertia weight $w$

$$w = w_{\max} - \frac{(w_{\max} - w_{\min})}{(1 + e^{-t/t_{\max}})} \tag{10}$$

It can be seen from Formula (10) that the change of inertia weight $w$ is nonlinear. In the beginning, when the value of $t$ is relatively small, the value of $w$ will become larger, which can speed up the global optimization speed. When the value of $t$ becomes larger later, the value of $w$ will become smaller to prevent the algorithm from crossing the boundary.

(2) Add adaptive mutation to particle swarm:

$$\text{rand} > \frac{1}{2} \left( 1 + \frac{t}{t_{\max}} \right) \tag{11}$$

From Formula (11), when iterations $t$ increases, the property of the inequality on the right will become larger as t grows, the probability of rand being larger than the right will also be smaller, and the probability of particle mutation will be lower so that PSO will avoid falling into the local optimal solution.

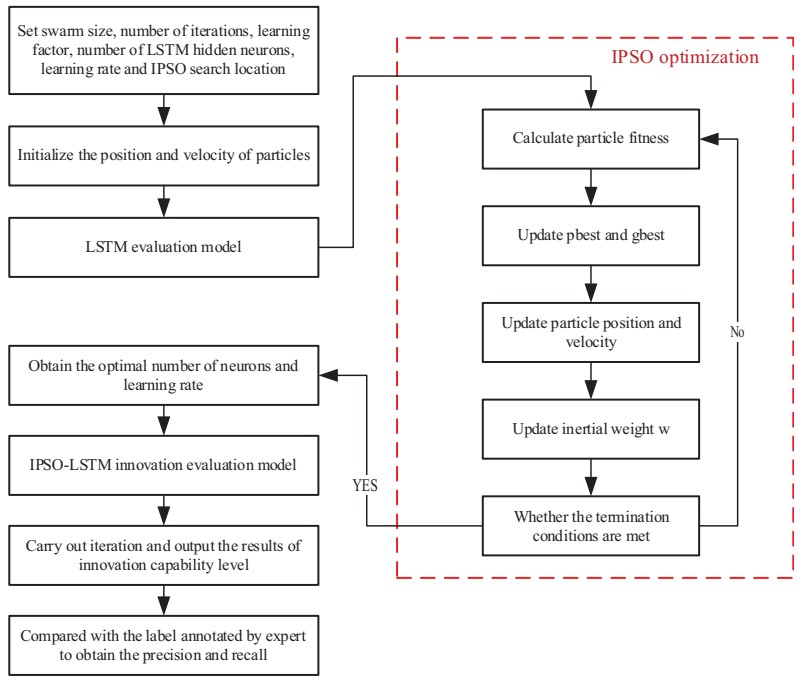

**Figure 2 The framework for the IPSO-LSTM.**

After completing the optimization of PSO, this article optimizes LSTM through IPSO method, and the overall process is shown in Fig. 2:

From the framework shown in Fig. 2, it can be seen that we first initialize the PSO method, which includes setting the population size, the maximum number of iterations, learning of the neural network, and the number of hidden layers. After completing the condition setting, initialize the positions and velocities of all particles and send them to the LSTM model for training. Train the model based on the set loss function and corresponding algorithm execution standards. Based on the training results, we obtained the corresponding data model to calculate precision, recall and other indicators according to the requirements.

## EXPERIMENTS AND ANALYSIS

The primary focus of this article revolves around the evaluation model of innovation capability based on students' psychological cognition levels. To assess students' psychological cognition level, the study draws inspiration from the characteristics of Jung's psychological test. It selects 20 highly relevant questions about personality traits and supplements the questionnaire with questions about innovative knowledge, resulting in 30 questions. Given the qualitative nature of these descriptions, which involve degree-based explanations, the Delphi method is employed for recording and quantification. For qualitative data, the Delphi method directly yields the corresponding membership degree. On the other hand, for quantitative data, a combination of the Delphi method and the membership function method is utilized to determine the membership function for each evaluation factor, thereby quantifying qualitative index data into numerical values.

Regarding the performance evaluation of the innovation capability model, the accuracy of data classification assumes utmost importance. Hence, this article employs three widely used evaluation indicators in machine learning: precision, recall, and F1-score. The specific calculation method is demonstrated in Formulas (12) to (14):

$$P = \frac{TP}{TP + FP} \tag{12}$$

$$R = \frac{TP}{TP + FN} \tag{13}$$

$$F1 = \frac{2 \times P \times R}{P + R} \tag{14}$$

where TP is the true positive, FP is the false positive and FN is the false negative.

## The model training of the IPSO-LSTM

Based on the mentioned results, 351 valid questionnaires were collected from students during the survey. The survey was conducted proportionally, with 40 students actively participating in innovation activities, 50 students taking part in school-organized innovation activities, and the remaining data being derived from teacher evaluations of the students.

For data analysis and model development, the author divided the data into an 8:1:1 ratio, where 80% of the data was used for training and the remaining 10% each for validation and testing purposes. To assess students' innovation ability, the author followed the suggestions of the innovation curriculum instructor and categorized the students into three grades: excellent, average, and failed. During the establishment of the model, the author calculated the results at different stages to facilitate a comprehensive analysis of the model's performance. The specific results are depicted in Fig. 3:

Figure 3 shows that the proposed method achieves a precision of 0.947 in the validation set test without adjusting the hyperparameters. However, the recall and F1-score exhibit considerable differences, indicating an imbalance in the model's performance.

To address this issue and achieve a more balanced model, this article conducts hyperparameter optimization during the resulting test of the validation set. The grid search method is employed in the optimization process to fine-tune the hyperparameters. After completing the hyperparameter optimization, the proposed method demonstrates a relatively balanced performance in the test set, effectively fulfilling the evaluation of students' innovation ability. This optimization ensures that the model's precision, recall, and F1-score are more harmonized and accurate in representing students' innovation capabilities.

## The model comparison using different optimization methods

This article employed the IPSO method to optimize the model and mitigate the risk of falling into local optimal solutions. To evaluate the model's performance, a comparison was conducted among three variants, LSTM, PSO-LSTM, and IPSO-LSTM. The loss function curve of these three methods during the training process is illustrated in Fig. 4:

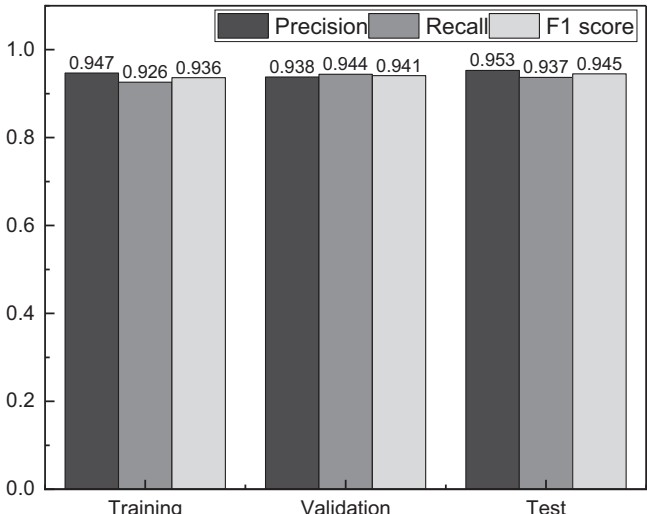

**Figure 3 The result for the student innovation ability evaluation using IPSO-LSTM.** The precision of the method proposed is 0.947 without adjusting the hyperparameter in the validation set test, but its recall and precision are quite different.

Figure 4 shows that IPSO-LSTM surpasses the other two methodologies, namely LSTM and PSO-LSTM, concerning the loss function index. IPSO-LSTM showcases a swifter convergence rate. Although all three approaches achieve a relatively stable convergence state by the 60th iteration, IPSO stands out by yielding markedly diminished loss values and superior evaluation performance. Additionally, this study juxtaposes the evaluation accuracy of the trio of methodologies, as illustrated in Fig. 5:

The Fig. 5 shows that the proposed IPSO-LSTM exhibits remarkable efficacy in evaluating innovation capability across various levels while ensuring minimal convergence speed and loss. Additionally, this article conducts a comparison with traditional machine learning methods, and the results are presented in Fig. 6:

Figure 6 shows that conventional machine learning methodologies, exemplified by LR, SVM, and DT, generally manifest precision levels hovering around 85%, with SVM attaining the zenith at 87.5%. Nevertheless, the overarching precision of these conventional methods remains inferior to that of RNN. Furthermore, the recognition accuracy of the singular RNN method also falls short compared to LSTM, accentuating LSTM's superior data processing prowess. This comparative analysis underscores the merits of adopting LSTM over traditional machine learning techniques to assess innovation capacity. LSTM's inherent capability to process sequential data and capture protracted dependencies makes it a potent instrument for undertakings.

## The results of the practical test

To conduct a more detailed evaluation of innovation capability based on students' psychological cognition level, the author selected five students from the pool of excellent students in the experiment for further analysis. During the analysis, the author examined the characteristic contribution rate of psychological and innovation indicators based on the design of the questionnaire. The specific results of this analysis are presented in Fig. 7:

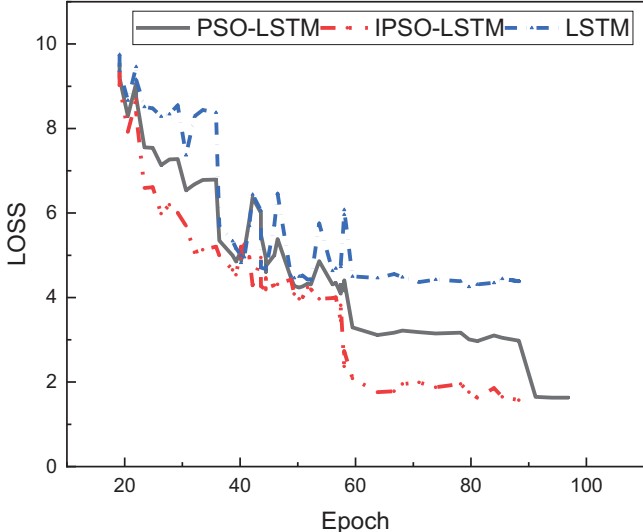

**Figure 4 The loss value among different methods.** The IPSO-LSTM presented performs better under the loss function index, and its bracelet speed is fast.

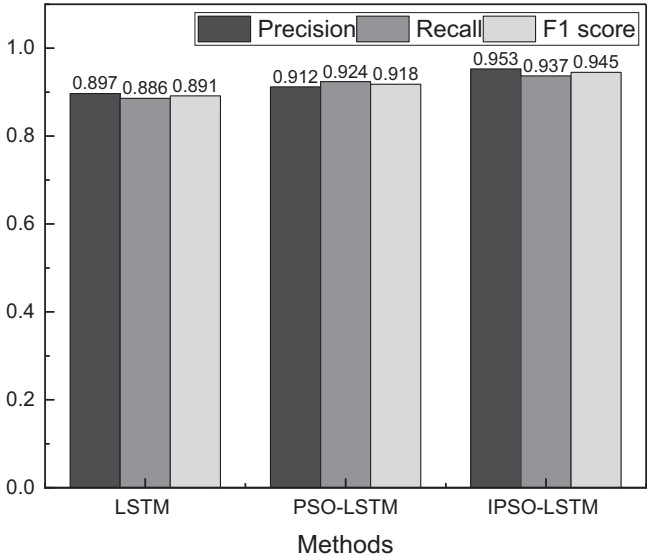

**Figure 5 The result of the evaluation using different methods.**

  

In Fig. 7, the results indicate that for the five students who were evaluated as excellent in innovation ability, the contribution rate of their psychological indicators is over 50%. This observation leads us to infer a significant correlation between students' innovation capability and their psychological qualities and personality traits. Notably, the psychological indicator for Student three demonstrates a contribution rate close to 70%, suggesting that this student possesses a high level of psychological cognition.

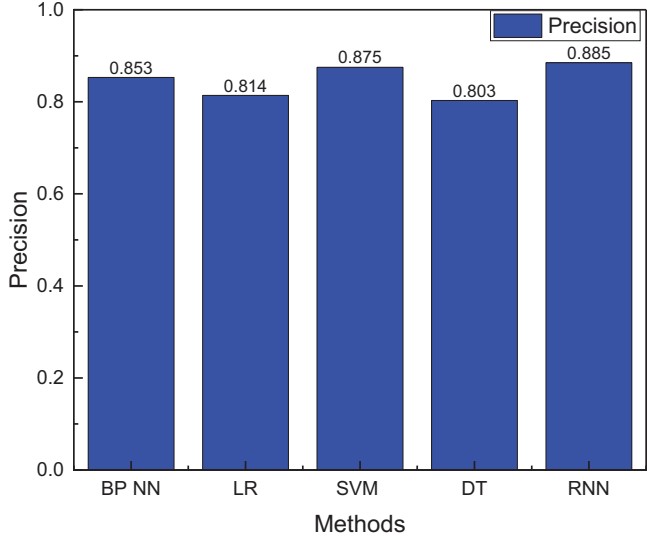

**Figure 6 The precision for the innovation evaluation using machine learning methods.**

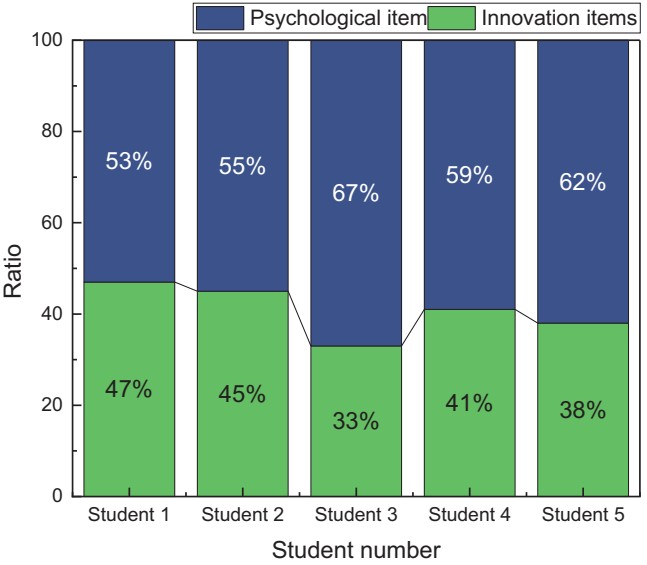

**Figure 7 The result for the feature contribution in the real test.**

These findings underscore the importance of students' psychological aspects of their innovation ability. In the context of future innovation education, it is recommended to strengthen students' psychological support and intervention appropriately. By focusing on psychological suggestions and interventions, the overall level of innovation education can be effectively enhanced, thereby fostering a conducive environment for nurturing innovative capabilities among students.

## DISCUSSION

Evaluating innovation ability has long been a prominent research area in higher education. This article innovatively incorporates psychological factors into the original survey and innovation assessment, thus formulating an evaluation model based on students' psychological cognitive level. To quantitatively analyze qualitative data, the article adopts the Delphi method, which significantly reduces the need for manual labour and improves accuracy when compared to traditional methods like AHP. Recognizing that innovation capability evaluation essentially entails a multi-objective regression optimization problem, the article chooses the LSTM method to ensure evaluation precision. To address the challenge of local extrema in neural network methods, the article proposes the IPSO method, optimizing particle weight and adaptive particle performance based on traditional PSO optimization. Experimental tests reveal the superiority of the IPSO method over PSO in terms of optimization performance. Moreover, the article conducts a characteristic contribution analysis of the model, scrutinizing the psychological factors and innovation evaluation characteristics in detail. This analysis demonstrates the crucial role of psychological factors in assessing innovation capability, providing valuable insights for future innovation education. This model can help educators better understand students' cognitive differences, facilitate personalized teaching and resource allocation, improve curriculum design pertinence, stimulate students' potential, and cultivate them as lifelong learners to adapt to constantly changing social and professional needs. This helps improve education quality, enabling students to better develop innovative abilities and cope with future challenges. This model can help educators achieve personalized teaching, better resource allocation, and potential stimulation. For policy makers, an innovative ability evaluation system established through students' psychological evaluation can help them formulate educational policies and propose more comprehensive evaluation models, thereby helping students achieve comprehensive development. After completing the preliminary establishment of the model, we need to conduct more data collection and analysis to provide practical and feasible visualization data and assist in evaluating innovation ability and implementing innovation education. At the same time, the model is only a means, and in implementation, it should fully consider the issue of teacher quality, enhance teacher training, monitor, and evaluate execution, and form a good feedback system. Only in this way can the model play its role.

The cultivation of college students' innovative psychological quality should extend beyond theoretical concepts and formalities; practical activities play a pivotal role in this process (*Chen & Li, 2022*). To foster innovative awareness among students, it is essential to incorporate innovation-focused elements in daily teaching, as well as in their study and daily life. Organizing innovation project activities and creating a campus environment that values and promotes innovation are equally crucial. Practical activities should be employed to stimulate students' competitiveness and foster their innovative spirit (*Palamarchuk, Gurevych & Maksymchuk, 2020*). Moreover, a close correlation exists between college students' mental health and innovative psychological quality. It is imperative that education for college students simultaneously encompasses mental health education and

creativity cultivation. Integrating mental health education into nurturing innovative psychological quality can complement and reinforce each other. Mental health education should be considered a fundamental component of college students' innovative psychological quality education (*Andersson, Titov & Dear, 2019*). Applying the theories of innovative psychological quality and college student's mental health, educational institutions should engage in diverse forms of innovative practice activities. These activities should be integrated into school education and management practices to foster positive optimism, a diligent learning attitude, lofty ideals, and goal-setting among college students. Innovation awareness and enthusiasm should be cultivated through activities stimulating students' competitiveness. While providing psychological counselling, universities should actively encourage and guide college students to engage in innovative activities. This multifaceted approach will enhance college students' mental health and innovative psychological quality.

## CONCLUSION

This article centres on evaluating students' innovative ability during higher education and introduces students' psychological cognition level into the assessment process from a psychological perspective. The Delphi method is utilized for data quantification to address the challenge of quantifying qualitative psychological cognitive data and making it applicable in deep learning machine learning models. Additionally, the article proposes the IPSO optimization method to optimize particle performance, thereby enhancing the optimization effectiveness. The experimental results demonstrate that the IPSO-LSTM method achieves an accuracy of 95.3%, surpassing both the single LSTM and PSO-LSTM methods in classification accuracy and model balance. Furthermore, to thoroughly assess the contribution of psychological and innovation indicators to students' innovation ability, the article conducts an analysis of students with excellent innovation capabilities. The results reveal that the contribution rate of psychological indicators to the characteristics of students with excellent innovation ability exceeds 50%, providing further evidence of the crucial role of students' psychological cognition level in fostering innovation capability.

Despite the notable achievements of this study, there are some limitations. The restricted number of invited experts limits the ability to expand the research sample comprehensively. To address this limitation and enhance the robustness of the model, future research should focus on expanding the sample size, increasing the number of psychological characteristics, and striving to improve the accuracy of the evaluation. By addressing these aspects, the evaluation model of innovation capability can be further refined and strengthened. For the evaluation of future innovation ability, more physiological signals and psychological tests should be integrated to conduct a more scientific evaluation of students' innovation ability from a psychological perspective, and more targeted curriculum should be designed based on the test results to cultivate students' innovation ability and related qualities. This is also a key direction for future application implementation.

### Funding

This work was supported by the Key Project of Guangdong Primary and Secondary School Teachers' Education and Scientific Research Ability Improvement in 2022 "Research on the Practice Path of Future School Construction Oriented to the Cultivation of Innovative Talents" (No: 2022ZQJK108). The funders had no role in study design, data collection and analysis, decision to publish, or preparation of the manuscript.

### Grant Disclosures

The following grant information was disclosed by the authors:
Key Project of Guangdong Primary and Secondary School Teachers' Education and Scientific Research Ability Improvement: 2022ZQJK108.

### Competing Interests

The authors declare that they have no competing interests.

### Author Contributions

- Fei Wan conceived and designed the experiments, performed the experiments, analyzed the data, performed the computation work, prepared figures and/or tables, authored or reviewed drafts of the article, and approved the final draft.

### Data Availability

The code is available in the Supplemental Files. The data is available at Zenodo: Dumas, Denis, Doherty, Michael, & Organisciak, Peter. (2020). Data from the "The Psychology of Professional and Student Actors: Creativity, Personality, and Motivation" (v.1.1.1) [Data set]. Zenodo. https://doi.org/10.5281/zenodo.3899579.

### Supplemental Information

Supplemental information for this article can be found online at http://dx.doi.org/10.7717/peerj-cs.1679#supplemental-information.

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
