# Peer review of "Design of innovation ability evaluation model based on IPSO-LSTM in intelligent teaching"

_PeerJ Computer Science, doi:10.7717/peerj-cs.1679_

## Round 0.1 · original submission · Major Revisions

Dear Authors,

Thank you for submitting your paper to PeerJ Computer Science. Please find the reviewers' feedback. Please response well to all given comments and consider the below main issues:

- The research problem should be described well in introduction, supported by up-to-date references. The list of contributions should be clearly listed.

- The methodology and the IPSO-LSTM validation should be described in more details.

- The performance of IPSO-LSTM model shoukd be compared with other state-of-the-art models  and discuss the findings well.

- DELPHI method should be described well. Please discuss the statistical analysis of your findings well.

Kind regards,

Faisal Saeed
Associate Editor

Reviewer 1 ·

Basic reporting

Aiming at the problem that the qualitative analysis of psychological, cognitive data is difficult to quantify and cannot be used in the deep learning model of machine learning, the Delphi method is used to quantify the data, and then an IPSO optimization method to optimize the performance of particles is proposed to improve the optimization effect. Experimental results show that the classification accuracy and model balance of IPSO-LSTM are higher than those of single LSTM and PSO-LSTM methods. I have the following concerns to improve the quality of the paper.

(1) Clearly outline your study's primary objectives and research questions to guide readers effectively.

(2) Expand the literature review to include more recent and relevant sources that support the link between innovative education, psychological states, and the knowledge economy.


(3) Offer a comprehensive description of the IPSO-LSTM model, including its architectural details and how it achieves high-precision evaluation and classification of innovation capabilities.

(4) Compare the performance of your IPSO-LSTM model with other state-of-the-art models used for innovation capability evaluation and discuss its advantages and limitations.


(5) Provide a more comprehensive analysis of the characteristic contribution results and their implications for policymakers, educators, and other stakeholders.


(6) Include a section that explicitly outlines the practical implications of your research and how it can inform educational policies and practices.

(7) Propose specific avenues for future research that can build upon your work and further enhance our understanding of the relationship between psychological states and innovation ability.

(8) In the discussion part, the author must mention the detailed comparison of results obtained in the present study with the previously published results.


I believe that addressing these suggestions will significantly enhance the quality and impact of your paper. Your work has the potential to contribute valuable insights to the fields of innovative education and psychological research, and I look forward to seeing the revised version.

Experimental design

Some sımulations are done. They should be improved.

Validity of the findings

They are good.

Additional comments

No comment

Reviewer 2 ·

Basic reporting

This work work presents valuable insights into the critical relationship between innovative education and psychological status. However, there are some aspects that require attention and improvement to strengthen the clarity and impact of your findings. Below, I provide detailed feedback and offer specific suggestions for revision:

Experimental design

1. In the introduction, provide a more comprehensive overview of the knowledge economy and the significance of innovative education in today's fiercely competitive world.

2. Elaborate on the DELPHI method: Provide a step-by-step explanation of how the qualitative description of psychological data was quantified using the DELPHI method to ensure reproducibility and rigor.

3. Discuss the methodology and the data used to validate the IPSO-LSTM model's classification accuracy and its applicability to different educational settings.

4. If applicable, discuss the ethical considerations involved in the study, particularly concerning the use of psychological data and its implications for students' privacy.

5. Elaborate on the sample used in the study and its representativeness, as well as any potential limitations in generalizing the results to broader student populations.

Validity of the findings

6. Consider discussing potential biases in data collection and analysis, and how they may impact the interpretation of the results.

7. Strengthen the conclusion by summarizing the key findings and emphasizing their significance in the context of innovative education and the knowledge economy.

8. Ensure the paper is well-written, coherent, and follows a logical flow, making it accessible to a broader readership.

9. In the discussion part, the author needs to mention the detail comparison of results obtained in the present study with the previously published results.

Additional comments

Please see the above sections for detailed comments to be incorporated.

Reviewer 3 ·

Basic reporting

optimize the abstract- reduce the introduction sentences at the beginning

this sentence "Teachers 52 who have received comprehensive innovation education are more likely to exhibit innovative
behaviors during their teaching practices. " in the introduction is a claim which needs a reference to support.

you have not elaborated the findings related to Delphi. for instance what was features set (indicators) you collected and intorduced to education experts. thoug, you used "Jung's psychological test" which means it is confimred, any justification why experts were needed?
based on what you determine this features set "Jung's psychological test", why only psychological with innovation indicators not other indicators? what was the result of 2nd round meeting with expert.

what was the feedback of experts (how many? you did not mention the number) regarding the "categorized the students into three grades: excellent,average, and failed", you only reported the feedback of "innovation curriculum instructor"

though you compare your LSTM-based model with many machine learning methods, you did not highlighted or justified avoiding CNN-based models. there is a possiblity they achieve the same performance of your model


you did not mention how did you tackle the small size of your dataset. you have data of less than 500 students.


your discussion part is full of recommendations there is no real discussion to show the implication/contribution of your work.

your selection of the students' sample was not clear, nothing about their demographics, level or even major

Experimental design

a lot of things missing.
you should fully explaining the process of DELPHI, how do you conducted
you should given information about the number of experts and their qualification

you should explain how do you tackle the small size of dataset

Validity of the findings

you should provide enough justification to not compare your model with CNN-based model

---

## Round 0.2 · accepted · Accept

The authors address all comments given by the reviewers. They proposed an innovation ability framework that integrates students' psychological state and innovation evaluation indicators.

Reviewer 1 ·

Basic reporting

It is done in the revised paper. It looks great.

Experimental design

It is done in the revised paper. It looks great.

Validity of the findings

It is done in the revised paper. It looks great.

Additional comments

It is done in the revised paper. It looks great.

Reviewer 2 ·

Basic reporting

This paper proposes an innovation ability evaluation model based on IPSO-LSTM in intelligent teaching. The paper has good potential for readers and is also revised according to my suggestions.

Experimental design

The experimental design is of sufficient quality and satisfaction level.

Validity of the findings

Findings are valid and justified

Additional comments

The paper has been revised according to my previous suggestions. Therefore, I recommend its acceptance.